# *Helicobacter pylori*-Induced Inflammation: Possible Factors Modulating the Risk of Gastric Cancer

**DOI:** 10.3390/pathogens10091099

**Published:** 2021-08-29

**Authors:** Sushil Kumar, Girijesh Kumar Patel, Uday C. Ghoshal

**Affiliations:** 1Department of Zoology, Deen Dayal Upadhyaya, Gorakhpur University, Gorakhpur 273009, India; 2Health Sciences Center, Department of Cell Biology and Biochemistry, Texas Tech University, 3601 4th Street, Lubbock, TX 79430, USA; 3Department of Gastroenterology, Sanjay Gandhi Postgraduate Institute of Medical Sciences, Lucknow 226014, India

**Keywords:** chronic inflammation, intestinal helminth, gastric carcinogenesis, *Helicobacter pylori*, gut microbiota, exosomes

## Abstract

Chronic inflammation and long-term tissue injury are related to many malignancies, including gastric cancer (GC). *Helicobacter pylori* (*H. pylori*), classified as a class I carcinogen, induces chronic superficial gastritis followed by gastric carcinogenesis. Despite a high prevalence of *H. pylori* infection, only about 1–3% of people infected with this bacterium develop GC worldwide. Furthermore, the development of chronic gastritis in some, but not all, *H. pylori*-infected subjects remains unexplained. These conflicting findings indicate that clinical outcomes of aggressive inflammation (atrophic gastritis) to gastric carcinogenesis are influenced by several other factors (in addition to *H. pylori* infection), such as gut microbiota, co-existence of intestinal helminths, dietary habits, and host genetic factors. This review has five goals: (1) to assess our current understanding of the process of *H. pylori*-triggered inflammation and gastric precursor lesions; (2) to present a hypothesis on risk modulation by the gut microbiota and infestation with intestinal helminths; (3) to identify the dietary behavior of the people at risk of GC; (4) to check the inflammation-related genetic polymorphisms and role of exosomes together with other factors as initiators of precancerous lesions and gastric carcinoma; and (5) finally, to conclude and suggest a new direction for future research.

## 1. Introduction

Gastric cancer (GC) is the most common cancer contributing to 5.5% of all new cases of cancers. Moreover, it is the fourth most lethal cancer, resulting in 7.7% of all deaths worldwide [1]. It has a poor prognosis, with a survival rate of less than 5 years among 80% of cases [2]. Although the incidence of GC has decreased, it remains a significant global health burden, with the highest burden in Asia [1]. *Helicobacter pylori (H. pylori)* infects over half of the world’s population, and this bacterium is the main cause of non-cardia GC [3,4,5]. However, geographical variations of bacterial virulence, age of acquisition of infection, host genetics, and environmental factors may lead to variation in the incidence of GC [6]. Paleo Correa’s hypothesis suggests that gastric carcinogenesis progresses in multiple stages and is caused by various factors. The histological cascade associated with GC is well characterized. It proceeds from normal mucosa infected with *H. pylori* to chronic active gastritis, atrophic gastritis (AG), intestinal metaplasia (complete at first, then incomplete later), dysplasia, hyperplasia, and adenocarcinoma [7,8,9,10].

A chronic inflammatory state is known to be critical for *H. pylori*-induced GC; the molecular mechanisms underlying how *H. pylori* communicate with gastric epithelial cells directly or indirectly to trigger gastric carcinogenesis remain unknown. GC exhibits a multi-factorial etiology. Moreover, owing to some paradoxical observations, it is now speculated that in addition to *H. pylori* infection, diet and host genetic factors are responsible for the progress from *H. pylori*-induced inflammation to GC [9,11,12]. A few recent studies also proposed that the increasing prevalence of autoimmune gastritis and dysbiosis of gut microbiota and increased use of antibiotics and acid suppressants, might have led to the variation in the risk of GC, primary gastric lymphoma (PGL), and neuroendocrine carcinoma [13,14,15]. Inter-individual variations on disease susceptibility of GC can also be influenced by other infections such as Epstein-Barr virus, intestinal helminths, and host genetic differences in cytokine genes [9,16,17]. Therefore, this review may give a new insight into geographical and inter-individual variation in inflammation-related precancerous lesions and mechanisms of how *H. pylori* infection might alter the disease. This may be helpful for preventive strategies in subjects with high risk for the development of GC, a potentially fatal malignancy, and may further givea new, more stringent concept for clinical research.

## 2. Mechanism of Chronic Inflammation and Multi-Step Sequel of Gastric Carcinogenesis

GC is an inflammation-associated carcinoma promoted by *H. pylori* infection, characterized by ongoing chronic gastritis, formation of metaplastic epithelia, and finally genetic instability in the gastric mucosal epithelium [18,19]. The relationship between chronic inflammation and cancer dates back to Virchow, who, in 1863, hypothesized that the origin of cancer was at the sites of chronic inflammation [20]. Many shreds of evidence proved that the inflammation might result from persistent mucosal or epithelial cell colonization by *H. pylori,* which may cause GC [11,21,22]. Persistent inflammation leads to increased cellular turnover, especially in the epithelium, and provides selection pressure, which may result in the emergence of cells at high risk for malignant transformation [23]. The association of inflammatory signals with intracellular pathways in gastric epithelial cells eventually leads to uncontrolled cell division, and differentiation remains inadequate. Whereas acute injury and inflammation associated with healing are usually self-limited, chronic injuries or inflammation over decade’s leads to a sustained expansion of proliferative tissue zones that are predisposed to neoplastic progression [24].

The precancerous cascade advances slowly and steadily, and it may take years or several decades to develop malignancy since the initiation of the cascade. However, the rate of progression from metaplasia to dysplasia is not the same in all individuals. A study revealed that the progression rate was two times higher in subjects more than 40 years of age than their younger counterparts [25]. In a subset of patients, this inflammatory process leads to loss of parietal cells and development of AG, followed by intestinal metaplasia (IM), dysplasia, and cancer [26]. Surprisingly, the majority of patients infected with *H. pylori* not only incline towards the development of pre-malignant lesions or GC, but also towards other gastroduodenal diseases (Figure 1). It is also speculated that all the stages before the development of dysplasia are reversible, although this is still somewhat controversial [24,25,26].

## 3. *H. pylori*-Triggered Inflammation and Gastric Precursor Lesions

### 3.1. The Link between H. pylori-Induced Inflammation and Gastric Carcinogenesis

The persistent inflammation caused by *H. pylori* recruits immune cells through steady generation of cytokines, resulting in histological changes, including pre-neoplastic gastric lesions (AG and IM), as well as the production of reactive radicals that might disrupt the host’s DNA, favoring the onset of carcinoma [27]. The International Agency for Research on Cancer (a branch of the World Health Organization) has previously approved the link between *H. pylori* and GC [28]. In the previous three decades, *H. pylori* has been implicated in the development of gastroduodenal diseases, including GC. Several case-control studies retrospectively examined the association between *H. pylori* and GC and revealed up to six-fold increased risk of GC in the presence of *H. pylori* infection [4,29,30,31]. The virulence of the infecting strain is a key predictor determining who would develop disease in people infected with *H. pylori*. Cytotoxin-associated antigen A is a marker for the Cag pathogenicity island, which contains genes required for pathogenic strains to cause increased inflammation [32,33].

Gastric inflammation not only results from *H. pylori* infection; dietary and environmental factors also influence the development of gastritis [34]. *H. pylori* (or its products) may produce gastric inflammation by two main mechanisms. Firstly, the organism may interact with epithelial cells on the surface, causing direct cell injury or the release of epithelial-derived pro-inflammatory mediators (chemokines). Secondly, *H. pylori*-derived products may obtain access to the mucosa under the surface, triggering non-specific and specific immune responses in the host, including the production of cytokine messengers. Thus, *H. pylori* infection induces both humoral and cellular immune responses.

*H. pylori*-induced inflammation, which includes neutrophil and macrophage activation and induction of Th1 response as well as alterations in host physiological responses associated with infection, contributes to mucosal damage [35]. Since *H. pylori* induces a strong humoral immune response, the Th2-cell response is expected. Paradoxically, *H. pylori*-specific stomach mucosal T-lymphocytes are mostly Th1 in nature [36]. This Th1 phenotype may be related to increased antral IL-18 production, and when combined with Fas-mediated apoptosis of *H. pylori*-specific T-cell clones, may favor *H. pylori* persistence [37,38]. Studies in gene-targeted mice have further shown that Th1 cytokines cause gastritis, but Th2 cytokines protect against inflammation [39]. The *H. pylori*-secreted peptidyl-prolyl cis, trans-isomerase, also drives the Th17 response, which promotes pro-inflammatory, low cytotoxic, gastric, tumor-infiltrating lymphocytes, matrix breakdown, and pro-angiogenic pathways, ultimately leading to the development of GC [40]. However, the enigma of why only a subset of *H. pylori*-infected individuals develops GC still continues.

### 3.2. The Link between H. pylori-Triggered AG, IM, and GC

Most individuals infected with *H. pylori* have infiltrating polymorphonuclear leucocytes in the antrum and the fundus of the stomach in the acute stage and lymphocytic inflammatory response later [41]. It is very slow to disappear after eradicating *H. pylori* [42]. In the treatment of peptic ulcers, early precancerous lesions of GC, and PGL, eradication of *H. pylori* infection is recommended [43,44]. In some cases, chronic gastritis is associated with gastric ulcers, which may increase the risk of GC, though duodenal ulcers caused by *H. pylori* are negatively associated with GC development [4,45,46].

Atrophic gastritis (AG), which is defined as the loss of specific glandular tissues, is a well-known early morphologic alteration that occurs before the onset of GC. However, the presence of AG is associated with a 0.5% to 1% annual risk of GC advancement [47,48]. Two types of AG exist in stomach mucosa: the first one is *H. pylori*-associated multifocal AG and the second form is corpus AG with anti-parietal cell and anti-intrinsic factor antibodies, which are associated with pernicious anemia. The process of AG occurs in approximately half of the *H. pylori*-colonized population, and it develops initially in those people and in those parts of the stomach where inflammation is the most severe [49,50]. In particular, *H. pylori* Cag+-pathogenicity-island-containing strains induce inflammation and have a higher risk of development of AG and GC [51,52,53,54]. Areas of gland loss and IM extend with time multi-focally, and although these do not give rise to any specific symptoms, these increase the risk for GC by 5- to 90-fold depending on the extent and severity of AG [55]. Paradoxically, patients with advanced AG and acholiorhydria revealed that even after eradication of *H. pylori*, there may be considerable risk of GC [56].

The link between *H. pylori*-triggered inflammation and IM is well established and is considered a major determinant of gastric carcinogenesis, but the bacterium is frequently undetected in these lesions [57,58]. Pre-neoplastic metaplasia has also evolved in *H. pylori*-infected mice models, notably spasmolytic polypeptide-expressing metaplasia (SPEM) [59]. IM occurs when the mucosa of the stomach mimics intestinal epithelium with goblet cells [60]. IM is classified into three types by Jass and Filipe, particularly type I (complete form), type II (incomplete form), and type III (intermediate form) [61]. The presence of type II or III IM is found to be related to a 20-fold greater risk of GC [62]. Some studies also proved the link between *H. pylori*-induced gastritis and IM [63,64]. Three mechanisms could generate this risk: (i) the metaplastic tissue is an early step in a multi-step induction process of gastric carcinogenesis; (ii) IM exhibits epigenetic changes, raises the pH of gastric juice by replacing oxyntic mucosa, and favors the growth of *H. pylori* capable of producing endogenous mutagens; (iii) IM is only a marker for chronic gastritis due to *H. pylori* infection or pernicious anemia [55,60]. Individuals with a high risk for GC can be identified using IM.

## 4. Risk Modulation of GC by Co-Existence of Gut Microbiota and Infestation with Intestinal Helminths

### 4.1. Gut Microbiota Other Than H. pylori and Their Role in Cascade of Gastric Cancer

Metagenomics and advanced nucleotide sequencing techniques unhooked the mucosal and luminal composition of the gut microbiota and revealed their significance in natural habitats other than *H. pylori* [65,66,67,68]. In transgenic mice, gastrointestinal over-expression of human IL-1β was sufficient for the development of gastric dysplasia and carcinoma in a stepwise manner [69,70]. Some experimental animal models, particularly transgenic insulin-gastrin (INS-GAS) mice model, showed the importance of non-*H. pylori* gastric microbiome in enhancing the effects of *H. pylori* in the development of GC [71,72]. Patients with GC also showed dysbiotic microbial population with genotoxic potential that differs from the patients with chronic gastritis [73]. The *H. pylori* infection generates an inflammatory reaction in the stomach, resulting in the loss of parietal cells and an elevation in gastric pH. *H. pylori* may contribute to microbial dysbiosis, and effective eradication can restore the gut microbiota to a state comparable to that of uninfected people [74]. The colonization of gut microbiota grows over time as AG develops, and in spite of *H. pylori* becoming diminished, the precancerous lesionscontinue to develop (Figure 2). Higher metabolites produced by dysbiosis of the gut microbiome, such as N-nitroso compounds and lactate, are thought to affect the immunological response as well as DNA damage, resulting in gastric carcinogenesis [75,76]. Recent studies proved that gut microbiota plays a significant role in the progression of gastric inflammation, AG, and IM after *H. pylori* eradication [77,78]. However, retrospective and only association-based findings are the limitations of these studies. It is unclear that the microbial changes seen in GC cause disease or are a result of the histologic progression through the precancerous cascade. Therefore, better understanding of the role of the gut microbiota in the development and progression of GC should lead to better diagnostic and preventive options.

### 4.2. Intestinal Helminth Infection Down-Regulates the Effect of H. pylori in Gastric Carcinogenesis

Co-infection with intestinal helminths may affect the outcome of *H. pylori* infection. A study on two Colombian communities infected with virulence-associated genotypes of *H. pylori* found a high versus low incidence of GC (endemic area of helminth infection), supporting the notion that intestinal helminth infection reduces the influence of *H. pylori* on gastric carcinogenesis [79]. Concurrent helminth infection has been shown in animal experiments to reduce the severity of *H. pylori*-induced gastritis [80]. Recently, a study on gastric mucosal samples also showed decreased expression of proinflammatory cytokines and predominant Th2 response (higher level IL-4) among *H. pylori*-infected humans co-infected with intestinal helminthes [81]. The higher load of *H. pylori* and intestinal parasites in the India and Venezuela populations is associated with a low risk of GC, which might explain these enigmas [9,81,82]. One study in a Chinese population found that concurrent helminth infections altered serological IgG responses as well as the pepsinogen I/II ratio, indicating a lower chance of developing *H. pylori*-induced atrophy [83]. A study on children from regions of low versus high risk of GC but similar *H. pylori* seropositivity showed that subjects from the low-risk area were more commonly infected with helminths and showed higher Th2-associated IgG1 responses to *H. pylori* infection [84]. These findings suggest that early childhood exposure to intestinal helminths induces immunoregulatory lymphocytes and anti-inflammatory cytokines such as IL-4, IL-10, and TGF-β and lowers the expression of pro-inflammatory IFN-γ, TNF-α, and IL-1β, including Th1-associated IP-10, RANTES, and MIP-1β. Cysteinyl leukotrienes are produced by tuft cells during helminth infection [85]. On the other hand, chemosensing by tuft cells activates group 2 innate lymphoid cells (ILC2), leading to an increase in tuft cell frequency, and exhibiting significant physiologic alterations in the tissue, including hyperplasia of mucus-secreting goblet cells [86]. Intestinal remodeling and helminth removal require this feedback control pathway. This mechanism provides a new direction for future research towards co-infection of helminths and *H. pylori*. Another study showed that *H. pylori* infection causes diseases by hypermethylation of key cellular promoters at CpG dinucleotides (promoter silencing), especially in gastric mucosa [87]. However, helminths may induce pathological changes by epigenetic reprogramming of host cells [88]. Poor hygienic and environmental conditions favor the endemic prerequisite for transmission and existence of both *H. pylori* and intestinal helminths acquired at an early age. In particular, helminths promote a Th2-polarizing response that may decrease *H. pylori*-induced cancer risk in individuals later in life (Figure 3). Studies addressing this issue are warranted and represent an intriguing area of research with the potential for future studies focused on *H. pylori* pathogenesis.

## 5. Relationship between Dietary Behavior of the People and Risk of GC

Diet may interact with bacterial and host genetic factors; dietary variations in different populations may explain difference in prevalence of GC in different areas of the world [89].Though gastric inflammation is caused primarily by *H. pylori* infection; exposure to other factors such as high salt, preserved food, and bile salts also influences the degree of gastritis [90,91,92,93]. Red and processed meat increase the oxidative stress to the inflamed stomach of individuals infected with *H. pylori*, putting them at higher risk of GC [94]. Many studies proved that smoking and eating a high-salt diet are both substantial risk factors for GC [95,96,97]. Obesity and alcohol (some say it is controversial depending on the amount) are also well-known risk factors for GC [98,99]. A diet rich in fresh fruits and vegetables, especially lycopene and lycopene-containing foods, and possibly vitamin C and selenium, can lower the risk of GC [100]. Many epidemiological studies found that diets high in antioxidant-rich fresh fruits and vegetables can minimize oxidative stress and lower the incidence of GC [101,102]. Turmeric has also been found to inhibit the growth of H. pylori and its anti-carcinogenic compound curcumin showed increased bioavailability in the presence of piperine (a compound from black pepper), and quercetine (a compound found in onion), which may decrease chronic inflammation [9,103]. In India, the eastern and southern parts of the country have a higher GC frequency than northern regions. Rice, non-vegetarian foods, mainly fish, are prevalent in the eastern Indian diet, which is spicy and with more salt [104]. In contrast, the northern Indian diet is wheat-based, and a greater proportion of people are vegetarian [104]. This observation suggests that geographical variations of dietary behavior are a significant component in the development and prevention of GC.

## 6. Inflammation-Related Genetic Polymorphisms Together with Other Factors as Initiators of Precancerous Lesions and GC

### 6.1. Cytokine Genes Influence Individual Response to Carcinogenic Exposures

Chronic inflammation can occur in genetically susceptible hosts with defective mucosal host defense systems or dysregulated immune responses, leading to excessively aggressive responses to ubiquitous antigens, which is the root cause of inflammation-related carcinogenesis [20,23,105]. Inflammation-related genetic polymorphisms act as initiators of *H. pylori*-induced chronic atrophic gastritis, and together with other factors they become the precursor of precancerous lesions and carcinoma [106,107]. Cytokines induced by specific stimuli, such as toxins produced by pathogens, are involved in immunity, inflammation, and cell proliferation. By secreting cytokines and recruiting specific inflammatory cells, *H. pylori* influence both the mucosal and systemic immune responses. In addition, *H. pylori* cause cellular changes as well as changes in genes that are important for epigenetic integrity and mucosal homeostasis. These genetic changes during the development of chronic inflammation are the subject of extensive investigation. A primary strategy for screening and early detection of GC in individuals at risk may include finding persons who have *H. pylori* infection with a pro-inflammatory makeup [108]. Genetic variations in pro-inflammatory and anti-inflammatory cytokine genes influence individual response to carcinogenic exposures. The degree of inflammation in the host tissue is determined not only by external factors such as infection with bacteria but also on the host’s genetic makeup, whether they have a high- or a low-producing genotype.

Pro- and anti-inflammatory cytokines modulate the inflammatory response of the stomach mucosa [109,110]. Pro-inflammatory cytokines activate inflammatory cells by the migration of neutrophils, mononuclear phagocytes, eosinophils, and mast cells (e.g., IL-8, MCP-1, RANTES, TNF-α,and IL-1), and also play a significant role in acquired immune responses that can regulate the growth, differentiation, and activation of lymphocytes, mast cells, eosinophils, and other hemopoietic cells (e.g., IFN- γ, IL-12 etc.) [111]. In another way, anti-inflammatory cytokine (e.g., IL-10, a product of Th2 cells) is a potent factor for suppressing the inflammatory and neoplastic environment. It inhibits IFN- γ production and antigen-specific T-cell activation by down-regulating antigen presentation as well as IL-1, TNF- α, and IL-6 production by monocytes or macrophages [112]. Cytokines, particularly SNPs of IL-1B, IL-4, IL-6, IL-8, IL-10, IL17A, and IL-17F, may influence cancer prognosis and prevention [113]. A balance between pro- (IL-8) and anti-inflammatory (IL-10) cytokines may influence the degree of chronic inflammation, which is a potential factor in development of gastritis and GC [16,114]. Furthermore, genetic diversity of cytokine genes revealed differences in the severity of *H. pylori*-induced inflammation and the risk of GC among various populations. More data will be necessary to assess the above hypothesis.

### 6.2. Inflammatory Gene Polymorphism Alters Acid Production

Acid production, oxidative stress, and DNA damage are all affected by polymorphisms in both bacterial and host genes, which are thought to be a major factor in the pathogenesis of GC [115]. When H. pylori infect the gastric mucosa, it stimulates neutrophils and mononuclear cells, causing them to release a variety of inflammatory cytokines [116]. In particular, IL-1β and TNF-α are potent inhibitors of gastric acid secretion [109,117,118].They promote the development of non-H. pylori gut microbiota capable of sustaining inflammation and continually producing oxidative stress, thereby hastening the process of gastric carcinogenesis. Moreover, long-term H. pylori infection in the context of vulnerable host gene polymorphisms may result in hypochlorhydria or hyperchlorhydria, depending on the degree and location of gastritis, which may proceed to gastric ulceration or GC. However, we believe the integration of all the information will be helpful for future study into novel molecular pathways and processes implicated in H. pylori-induced inflammation.

## 7. Role of Exosomes in Gastric Cancer

Exosomes are nanovesicles (30–150 nm) of endocytic origin and are largely recognized in the intercellular communications via transfer of vital contents, which may induce tumor cell proliferation, metastasis, inflammation, immune suppression, remodeling of tumor microenvironment, drug resistance, etc [119,120]. Chronic inflammation is one of the most common features associated with peptic ulceration and GC development and progression. Cells of tumor microenvironment are known to play a major role in chronic inflammation. It is reported that *H. pylori*-infected, GC-cells-derived exosomes were enriched with the phosphorylated (active) form of mesenchymal epithelial transition factor (MET) (Figure 4).Exosome-mediated transfer of MET modulates the macrophages towards a pro-tumorigenesis phenotype, stimulating the secretion of pro-inflammatory cytokine IL-1β that promotes GC cell growth and proliferation [121]. In another important finding, CagA, encoded by cytotoxin-associated gene A, a major virulence factor of *H. pylori,* was shown to be present in the exosomes isolated from the blood of *H. pylori*-infected patients [122]. These CagA positive exosomes modulate the gastric epithelial cell behavior. Circulating CagA positive exosomes secreted from gastric epithelial cells may be involved in disease spread and may help in the development of extra-gastric disorders associated with CagA-positive *H. pylori* infection [122]. SPEM is an important risk factor driven by macrophage-mediated chronic inflammation. It is demonstrated that exosomes derived from the macrophage induced by deoxycholic acid (a type of harmful secondary bile acid) increased the expression of SPEM markers (TFF2 and GSII lectin) in gastric organoids, indicating the role of macrophage-derived exosomes in the onsets of GC [123]. Zheng et al. (2017) showed that tumor-associated macrophage (TAM)-derived exosome enriched with miRNA-21 down-regulates PTEN, suppresses cell apoptosis, and enhances the activation of PI3K/AKT signaling pathway [124]. In another study, TAM-derived exosomes were found to be enriched with apolipoprotein E (ApoE) and were shown to activate PI3K-Akt signaling pathway in recipient GC cells and affect cytoskeleton remodeling and promote GC cells migration [124]. Similarly, GC cells-derived exosomes were shown to activate PI3K/Akt and mitogen-activated protein kinase/extracellular-regulated protein kinase pathways leading to tumor cell proliferation [125]. In another study, exosome derived from GC cells was shown to activate macrophages by inducing NF-κB activity and thus enhanced the release of pro-inflammatory cytokines, including IL-6 and TNF-α, which in turn promotes proliferation and migration of GC cells [126]. The expression of ZFAS1, a long noncoding RNA (lncRNA), is known to be elevated in the GC cells, tumor tissues, serum, and serum exosomes of GC patients. Mechanistically, ZFAS1 promotes proliferation and migration and suppresses apoptosis in GC cells [127]. Wei et al. (2020) reported that miR-15b-3p transferred via exosomes promotes GC development and GES-1 cell malignant transformation by inhibiting the expression of DYNLT1, Cleaved Caspase-9, and Caspase-3 [128].

Cancer-associated fibroblasts (CAFs) play a multifaceted role in cancer progression. It is shown that exosomes derived from the GC cells promote differentiation of mesenchymal stem cells (MSCs) into CAFs. The PKM2-enriched exosomes derived from the GC cells are associated with nuclear factor kappa B (NF-κB) activity and elevated levels of IL-6, IL-8, and other inflammatory cytokines [129]. CAFs are also known to play an important role in drug resistance. It is shown that CAF-derived exosomes transfer miR-522 to GC cells, suppressing the expression of ALOX15 and thus decreasing the lipid-ROS accumulation in GC cells, leading to resistance to paclitaxel and cisplatinm [130]. In a study by Zhang et al. (2018), GC-cell-derived exosomes, by transferring high mobility group box-1 (HMGB1), are shown to activate NF-κB signaling pathway. HMGB1 were shown to induce autophagy and activation of neutrophils via HMGB1/TLR4/NF-κB signaling axis and help in the progression of GC [131]. Altogether, these findings affirm the role of exosomes in chronic inflammation and growth and development of GC.

## 8. Conclusions and Future Directions

*H. pylori*: though it is known to be a carcinogen, other factors also influence its carcinogenic potential. The variation in the other factors may partly explain differences in the prevalence of GC in different populations in the world despite high frequency of *H. pylori* infection in some countries. These factors include the genetic factors of the host, diet, gut microbiota, and other infective agents such as helminths. The knowledge of the modifiable factors in the development of GC such as diet and gut microbiota, which are known to be modified with dietary change, may help in prevention of gastric carcinogenesis. It is critical to learn more about the pathogenesis of *H. pylori*-induced inflammation, as well as potential variables that influence the risk of GC, not just to find more effective therapies for this frequent malignancy, but also because it might serve as a model for how persistent inflammation plays a role in the development of other gastrointestinal malignancies. Moreover, knowledge about multifactorial pathogenesis of *H. pylori*-induced gastric carcinogenesis may help to predict the risk of development of GC in an individual with *H. pylori* infection by complex mathematical modeling; this may help in deciding on surveillance strategy, *H. pylori* eradication, testing for re-infection, and screening endoscopy, as mass eradication of *H. pylori* in an unselected population, even in high-risk areas, was not found to be rewarding.

## Figures and Tables

**Figure 1 pathogens-10-01099-f001:**
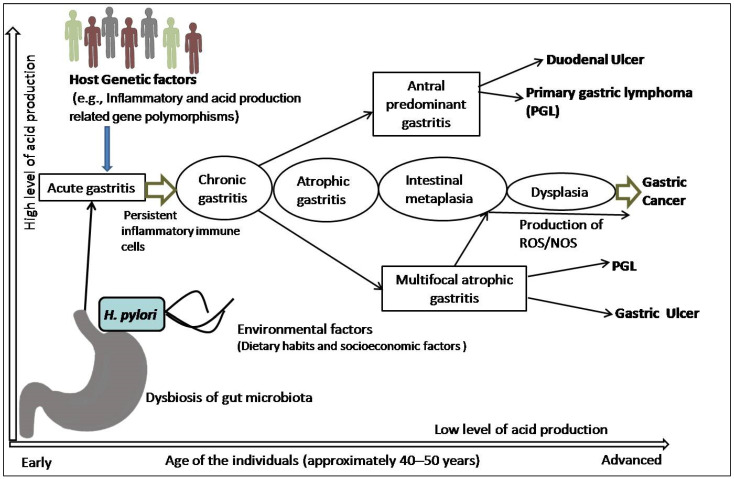
Schematic representation of pathophysiology of gastric cancer: *H. pylori* infection and the combination of inflammatory and acid-production-related genetic polymorphisms, including dietary and socioeconomic variables, lead to development of acute gastritis that subsequently becomes chronic gastritis in the presence of persistent inflammatory immune cells. In a subset of patients, this chronic inflammation with antral and corpus localization develops peptic (duodenal and gastric) ulcer or primary gastric lymphoma (PGL). However, dysbiosis, interactions with other variables, and the generation of reactive oxygen species (ROS)/nitrogen oxygen species (NOS) cause atrophic gastritis, which is followed by intestinal metaplasia, dysplasia, and carcinoma.

**Figure 2 pathogens-10-01099-f002:**
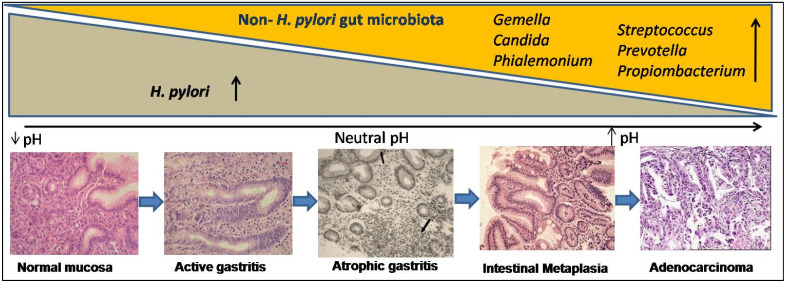
Gastric microbial dysbiosis process:in the histological course of gastric carcinogenesis (at the bottom), *H. pylori*-dependent and independent phases showed exponential changes in gut microbiota (above)*. H. pylori*-infected gastric mucosa shows low pH and after years to decades of colonization, *H. pylori* disappear and progressive atrophic gastritis, which reduces gastric acid secretion, raising the intra-gastric pH, favors overgrowing of non-*H. pylori* microbiota (i.e., *Gemella, Candida, Streptococcus, Prevotella, Propiombacterium, and Phialemonium),* which have carcinogenic potential.

**Figure 3 pathogens-10-01099-f003:**
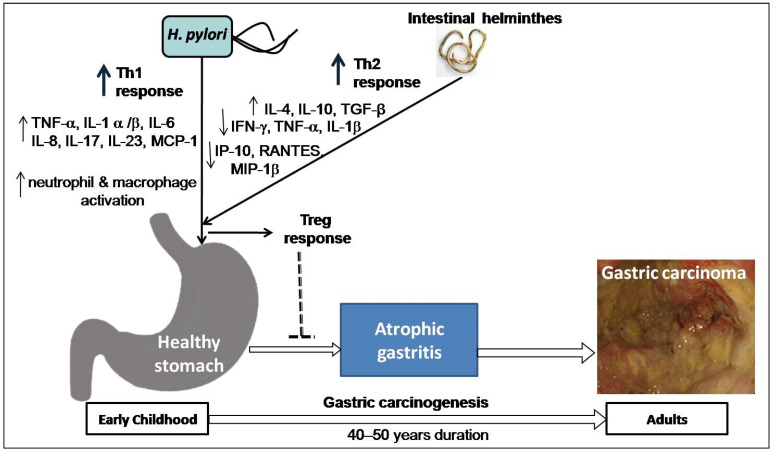
Hypothesis on the impact of co-existence of *H. pylori* and intestinal helminths: both *H. pylori* and intestinal helminth infection interact in early childhood, resulting in change in Th1 response caused by *H. pylori* primarily upregulation of tumor necrosis factor (TNF)- α, interleukin (IL)-1 α/β, IL-6, IL-8, IL-17, IL-23 and monocyte chemoattractant protein (MCP-1) as well as activation of neutrophil and macrophage and Th2-polarizing response by helminthes mainly upregulation of IL-4, IL-10, Transforming growth factor (TGF)-β and downregulation of interferon (IFN)-γ, TNF-α, and IL-1β, including Th1-associated interferon gamma-induced protein 10 (IP-10), Regulated on Activation, Normal T Expressed and Secreted (RANTES) protein, and Macrophage inflammatory protein (MIP)-1β. Thus, T regulatory (Treg) response inhibits the development of atrophic gastritis and possibly lowers cancer risk in later life (after 40–50 years).

**Figure 4 pathogens-10-01099-f004:**
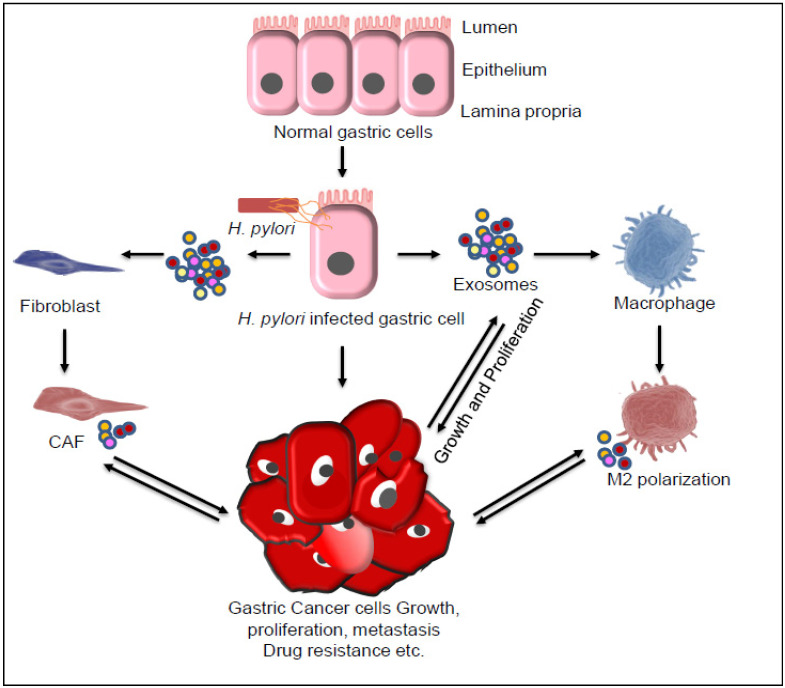
Exosome-mediated inter-cellular communications lead to gastric cancer (GC) progression. Different genetic and epigenetic factors and *H. pylori* infection alone or in combination lead to development of GC. Exosome derived from *H.pylori* or from infected gastric cells, cancer-associated fibroblasts (CAF), and macrophage marker (M2 macrophage) cells plays an important role in GC progression through different pathways by transfer of vital molecules, including Cytotoxin-associated antigen A (CagA), mesenchymal epithelial transition factor (MET), TFF2 and GSII ApoE (biomarkers of spasmolytic polypeptide-expressing metaplasia [SPEM]), high mobility group box-1 (HMGB1), pyruvate kinase M2 (PKM2), long noncoding RNA, and microRNA. Besides exosomes, the secreted inflammatory cytokines, such as interleukin (IL)-6, IL-8, and IL-1β also play a major role in cancer cell growth proliferation, metastasis, and drug resistance.

## Data Availability

Not Applicable.

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
