# Peer review of "Helicobacter pylori-Induced Inflammation: Possible Factors Modulating the Risk of Gastric Cancer"

_pathogens, 2021, doi:10.3390/pathogens10091099_

Round 1

Reviewer 1 Report

Brief summary

In this review, the authors have listed possible factors modulating the risk of cancer. In addition to the well-known risk factors H. pylori, the authors focus on gut microbiota, gut parasites, diet, gene polymorphisms, and exosomes to suggest new directions for future research.

Broad comments

Overall, the authors presented an organized review, however the coverage is too wide, which makes the content shallow.

The content is biased towards personal interests and the issue is not clear.

Specific comments

 I have encountered some shortcomings which have to be addressed.

1.On page 1-line40, you have written “intestinal metaplasia(complete at first, then incomplete later)”. There is no such description that intestinal metaplasia develops to incomplete type in the cited references.

2.Numbering of the paragraphs are not correct (page2,line 58 and page3,line 92,)

3.On page 4,line135 to 137, you have mentioned autoimmune gastritis, but not neuroendocrine carcinoma which is also a gastric cancer.

  1. Axis units should be listed outside the table in figure2.

5.On page 5,line164-165, you have mentioned INS-GAS mice model, which showed that Th2 immune responses and helminths associated with reduced progression. Since IL-1β overexpressing gastric cancer mouse model is well known, consider mentioning on page8,paragraph 6.2.  

6.The magnification of hematoxylin and eosin stain picture differs in Figure2.

7.Helminth infection is known to cause Tuft–ILC2 Circuit reaction driving goblet cell hyperplasia, and worm clearance in intestine but there are only few reports of gastric ILC2. Consider mentioning as a new direction for future research.

  1. If you write about infections and gastric cancer, you should also mention EB virus.
  2. In page 7,line 263 you write “ce+=lls”.You should write “cells” instead.
  3. In paragraph 5, you have mentioned dietary behavior. Obesity and alcohols(some say its controversial depending on the amount) are also well known risk factors for gastric cancer.
  4. Polymorphism and Exosomes are major topics, and only some of the knowledge in this area is covered in this review. The relationship with the pathogen is also unknown.
  5. Bold and thin characters are mixed in the text.

Author Response

Reviewer: 1

Broad Comments: In this review, the authors have listed possible factors modulating the risk of cancer. In addition to the well-known risk factors H. pylori, the authors focus on gut microbiota, gut parasites, diet, gene polymorphisms, and exosomes to suggest new directions for future research.

Overall, the authors presented an organized review, however the coverage is too wide, which makes the content shallow.

The content is biased towards personal interests and the issue is not clear.

Response: We greatly appreciate the encouraging notes regarding the organized review covering the broad area. Based on this special issue and our research interest, we covered the Helicobacter pylori-induced inflammation which is potential risk factor of gastric cancer along with other possible factors. Other groups may cover other factors in this special issue.    

Specific comments

Comment 1: On page 1-line40, you have written “intestinal metaplasia (complete at first, then incomplete later)”. There is no such description that intestinal metaplasia develops to incomplete type in the cited references.

Response: Thank you for this suggestion. Now reference has been added in page 1-line 13, “intestinal metaplasia (complete at first, then incomplete later)” Ref: 9 (Correa P, Piazuelo MB, Wilson KT. Pathology of gastric intestinal metaplasia: clinical implications. Am J Gastroenterol. 2010;105(3):493-498. doi:10.1038/ajg.2009.7282).

Comment 2: Numbering of the paragraphs are not correct (page2, line 58 and page3, line 92,)

Response: Thank you very much for pointing out the typos. We have incorporated the suggested corrections.

Comment 3: On page 4, line135 to 137, you have mentioned autoimmune gastritis, but not neuroendocrine carcinoma which is also a gastric cancer.

Response: Thank you for this suggestion. We have added the suggested statement about neuroendocrine carcinoma and added the references. (Terao, S., Suzuki, S., Yaita, H., Kurahara, K., Shunto, J., Furuta, T., Maruyama, Y., Ito, M., Kamada, T., Aoki, R., Inoue, K., Manabe, N. and Haruma, K. (2020), Multicenter study of autoimmune gastritis in Japan: Clinical and endoscopic characteristics. Digestive Endoscopy, 32: 364-372. https://doi.org/10.1111/den.13500)

Comment 4: Axis units should be listed outside the table in Figure 2.

Response: We have inserted the suggested correction. “Fig. 2: Gastric microbial dysbiosis process: In the histological course of gastric carcinogenesis (at the bottom), H. pylori dependent and independent phases showed exponential changes in gut microbiota (at above). H. pylori-infected gastric mucosa shows low pH and after years to decades of colonization, H. pylori become disappear and progressively atrophic gastritis stage with neutral pH becomes favorable for overgrowing of Non-H. pylori gut microbiota (i.e. Gemella, Candida, Streptococcus, Prevotella, Propiombacterium, and  Phialemonium), may produce carcinogenic potential.”

Comment 5: On page 5, line164-165, you have mentioned INS-GAS mice model, which showed that Th2 immune responses and helminths associated with reduced progression. Since IL-1β overexpressing gastric cancer mouse model is well known, consider mentioning on page8, paragraph 6.2.

Response: We have now revised the manuscript and added the suggested statement as per your recommendation.

“In transgenic mice, gastrointestinal over-expression of human IL-1β was sufficient for the development of gastric dysplasia and carcinoma in a stepwise (ref ….).”

Ref: Shuiping Tu, Govind Bhagat, Guanglin Cui, et. Al. Overexpression of Interleukin-1β Induces Gastric Inflammation and Cancer and Mobilizes Myeloid-Derived Suppressor Cells in Mice. Cancer Cell, 2008: 14 ( 5), 408-419.

Ref: Huang FY, Chan AO, Rashid A, Wong DK, Seto WK, Cho CH, Lai CL and Yuen MF: Interleukin‑1β increases the risk of gastric cancer through induction of aberrant DNA methylation in a mouse model. Oncol Lett 11: 2919-2924, 2016.

Comment 6: The magnification of hematoxylin and eosin stain picture differs in Figure2

Response: We have fixed the magnification for the images in Figure 2, now all the picture magnification of hematoxylin and eosin stain is X400 magnification.

Comment 7: Helminth infection is known to cause Tuft–ILC2 Circuit reaction driving goblet cell hyperplasia, and worm clearance in intestine but there are only few reports of gastric ILC2. Consider mentioning as a new direction for future research.

Response: Now we have added in page 10 helminth section “Cysteinyl leukotrienes are produced by tuft cells during helminth infection (ref). On the other hand, chemosensing by tuft cells, activates group 2 innate lymphoid cells (ILC2), leading to an increase in tuft cell frequency, exhibited significant physiologic alterations in the tissue, including hyperplasia of mucus-secreting goblet cells (ref). Intestinal remodelling and helminth removal require this feedback control pathway. This mechanism provides a new direction for future research towards co-infection of helminths and H. pylori.”

Ref 1: John W. McGinty, Hung-An Ting, et.al. Tuft-Cell-Derived Leukotrienes Drive Rapid Anti-helminth Immunity in the Small Intestine but Are Dispensable for Anti-protist Immunity. Immunity, 2020: 52( 3), 528-541.

Ref 2: Von Moltke J, Ji M, Liang HE, Locksley RM. Tuft-cell-derived IL-25 regulates an intestinal ILC2-epithelial response circuit. Nature.2016;529(7585):221-225. doi:10.1038/nature16161

Comment 8: If you write about infections and gastric cancer, you should also mention EB virus.

Response: As per the manuscript title “Helicobacter pylori-induced inflammation: Possible factors modulating the risk of gastric cancer” we are mainly fouccused on Helicobacter pylori-mediated induced inflammation and passible factor for GC. However, as suggested by the reviewer, in the revised manuscript we did mention a role of EB virus as well. We also added a reference supporting that.

Comment 9: In page 7, line 263 you write “ce+=lls”. You should write “cells” instead.

Response: Typos is corrected in the revised manuscript.

Comment 10: In paragraph 5, you have mentioned dietary behavior. Obesity and alcohols (some say its controversial depending on the amount) are also well-known risk factors for gastric cancer.

Response: Thanks for your suggestion. Now we have added these important risk factors. “Obesity and alcohols (some say its controversial depending on the amount) are also well known risk factors for gastric cancer (ref)”.

Ref 1: Yi Chen, Lingxiao Liu, Xiaolin Wang et.al. Body Mass Index and Risk of Gastric Cancer: A Meta-analysis of a Population with More Than Ten Million from 24 Prospective Studies. Cancer Epidemiol Biomarkers Prev August 1 2013 (22) (8) 1395-1408.

Ref 2. López-Carrillo, Lizbeth et al. Alcohol consumption and gastric cancer in Mexico. Cadernos de Saúde Pública [online]. 1998, 14,(3), S25-S32.

Comment 11: Polymorphism and Exosomes are major topics, and only some of the knowledge in this area is covered in this review. The relationship with the pathogen is also unknown.

Response: We agree with reviewer’s suggestion however, the coverage of exosome itself will be a full length of review article. Here our focus is H. Pylori-mediated inflammation and carcinogenesis where exosomes paly an important role besides other factors.

Comment 12: Bold and thin characters are mixed in the text.

Response: Thank you very much for pointing out the typos. We have unified the fonts in the revised manuscript.

Reviewer 2 Report

This review about H. pylori-induced inflammation related with GC development is well written. Some comments are below.

The authors described H. pylori-induced AG and IM in 3.2. section, but the description of SPEM is insufficient. They mentioned SPEM only in the role of exosomes. 

The authors mainly described inflammatory responses against H. pylori infection, but the role of H. pylori virulence factors, including CagA should be described additionally.

"precancerous lesions including carcinoma" in line 242 of page 7 should be changed. Precancerous lesions and carcinoma are different. 

The numbers of sections were incorrect. For example, "1" in line 58 of page 2 and "1" in line 92 of page 3 were incorrect. 

Some bold letters are inappropriate. For example, almost all sentences in the sections of 5 and 6.2. are bold. They should use bold letters only in most important sentences.

"ce+-lls" in line 263 of page 7 is incorrect.

"NF-kB" in line 343 of page 9 is incorrect.

Author Response

Reviewer 2

General Comment: This review about H. pylori-induced inflammation related with GC development is well written.

Response: We greatly appreciate encouraging comments of the reviewer.

Comment 1: The authors described H. pylori-induced AG and IM in 3.2. section, but the description of SPEM is insufficient. They mentioned SPEM only in the role of exosomes.

Response: Thank you for this suggestion. We have added description about SPEM on page in section 3.2. “Preneoplastic metaplasia has also evolved in H. pylori-infected mice models, notably spasmolytic polypeptide-expressing metaplasia (SPEM) ref.

Ref: Weis VG, Goldenring JR. Current understanding of SPEM and its standing in the preneoplastic process. Gastric Cancer 2009;12(4):189-197.

Comment 2: The authors mainly described inflammatory responses against H. pylori infection, but the role of H. pylori virulence factors, including CagA should be described additionally.

Response: Thank you for this suggestion. We have added the information pertaining to the functional role of CagA in section 3.1. Moreover, the role of CagA is described in section 7 and figure 4 in detail.

“The virulence of the infecting strain is a key predictor about who suffers disease in people infected with H. pylori. Cytotoxin-associated antigen A (CagA) is a marker for the cag pathogenicity island, which contains genes required for pathogenic strains to cause increased inflammation (ref)”

Ref 1: John C Atherton, H. pylori virulence factors, British Medical Bulletin 1998, Volume 54, Issue 1, Pages 105–120.

Ref 2: Graham, D.Y. and Yamaoka, Y. (2000), Disease-specific Helicobacter pylori Virulence Factors: The Unfulfilled Promise. Helicobacter, 5: 3-9.

Comment 3: "precancerous lesions including carcinoma" in line 242 of page 7 should be changed. Precancerous lesions and carcinoma are different.

Response: Thank you for reviewer’s suggestion. Correction has been incorporated in the revised manuscript. I have replaced “precancerous lesions including carcinoma" to “precancerous lesions and carcinoma”.

Comment 4: The numbers of sections were incorrect. For example, "1" in line 58 of page 2 and "1" in line 92 of page 3 were incorrect.

Response: Thank you for pointing out the numbering error. Correction has been incorporated in the revised manuscript.

Comment 5: Some bold letters are inappropriate. For example, almost all sentences in the sections of 5 and 6.2. are bold. They should use bold letters only in most important sentences.

Response: We unified the font of letters in the revised manuscript.

Comment 6: "ce+-lls" in line 263 of page 7 is incorrect.

Response: We corrected the typos in the revised manuscript.

Comment 7: "NF-kB" in line 343 of page 9 is incorrect.

Response: We corrected it in the revised manuscript as “nuclear factor kappa B (NF-κB)” mentioned in page 9 section 7.

We believe that we have satisfactorily responded to the reviewers’ comments. I will be glad to provide you any additional information, should you need it.

Round 2

Reviewer 1 Report

I find the author have responsed to all comments appropreately.

Reviewer 2 Report

The authors responded completely.